# Bioengineered bacterial vesicles as biological nano-heaters for optoacoustic imaging

Vipul Gujrati[1,2], Jaya Prakash[2], Jaber Malekzadeh-Najafabadi[1,2], Andre Stiel [2], Uwe Klemm[2],
Gabriele Mettenleiter[3], Michaela Aichler[3], Axel Walch[3] & Vasilis Ntziachristos[1,2]

Advances in genetic engineering have enabled the use of bacterial outer membrane vesicles (OMVs) to deliver vaccines, drugs and immunotherapy agents, as a strategy to circumvent biocompatibility and large-scale production issues associated with synthetic nanomaterials. We investigate bioengineered OMVs for contrast enhancement in optoacoustic (photo-acoustic) imaging. We produce OMVs encapsulating biopolymer-melanin (OMV$^{Mel}$) using a bacterial strain expressing a tyrosinase transgene. Our results show that upon near-infrared light irradiation, OMV$^{Mel}$ generates strong optoacoustic signals appropriate for imaging applications. In addition, we show that OMV$^{Mel}$ builds up intense heat from the absorbed laser energy and mediates photothermal effects both in vitro and in vivo. Using multispectral optoacoustic tomography, we noninvasively monitor the spatio-temporal, tumour-associated OMV$^{Mel}$ distribution in vivo. This work points to the use of bioengineered vesicles as potent alternatives to synthetic particles more commonly employed for optoacoustic imaging, with the potential to enable both image enhancement and photothermal applications.

[1] Chair of Biological Imaging, TranslaTUM, Technische Universität München, Munich 81675, Germany. [2] Institute of Biological and Medical Imaging, Helmholtz Zentrum München, Neuherberg 85764, Germany. [3] Research Unit Analytical Pathology, Helmholtz Zentrum München, Neuherberg 85764, Germany. These authors contributed equally: Vipul Gujrati, Jaya Prakash. Correspondence and requests for materials should be addressed to V.N. (email: v.ntziachristos@tum.de)

A broad range of synthetic nanoparticles made up of inorganic and organic materials such as quantum dots, Au, Ag, Cu and polymeric particles have been reported for diagnostic and therapeutic functionality[1–8]. Despite the appreciable success of synthetic nanomaterials for efficient disease diagnosis and therapy in preclinical trials, only a few synthetic agents have entered clinical trials. Factors that limit clinical dissemination of most synthetic nanomaterials include challenges involving low biocompatibility, material-associated toxicity, poor clearance and high cost of pilot scale production[9,10]. To overcome the technical limitations associated with synthetic nanomaterials, researchers in the last decade have shown keen interest in the development of cell-derived, nano-sized vesicles as carrier systems[11]. Moreover, bioengineering and bioprocessing tools have provided scalable and robust platforms for manufacturing such cells and vesicles to meet clinical and commercial needs.

Cell membrane-derived nano-vesicles have been explored with prokaryotic and eukaryotic cells, including mammalian cell-derived exosomes[12], erythrocyte-derived camouflaged particles[13], yeast vacuoles[14], bacterially derived minicells[15] and outer membrane vesicles (OMVs)[16,17]. Exosome-based systems show low stability and yield and require expensive purification and production methods. Erythrocyte-derived particles exhibit excellent biocompatibility because they are taken from the target organism, but their lack of a nucleus means that they cannot be genetically engineered to carry biologically derived cargo. Yeast vacuoles show good ability to penetrate tissues and are easy to scale up, but their long-term stability, immunogenicity and toxicity remain to be clarified. Bacterial OMVs show several advantages as nano-carriers since they attain a rigid membrane, which imparts stability and reduces leakage in systemic circulation. Moreover, OMVs are safe because they are acellular and can be used in vivo in very small quantities. Importantly, OMVs can be customised to carry desired payloads and can easily be produced in large quantities using fermentation and purification procedures previously optimised on a pilot scale. This is particularly beneficial when considering that bacteria can easily be modified genetically to produce desired agents useful in vaccination, bio-sensing, bio-imaging, therapy or targeted delivery; and these agents can be localised specifically in membrane-derived vesicles[16–20].

In this work we consider OMVs a platform for optoacoustic applications, which can be employed for contrast-enhancement and therapeutic applications. Multi-spectral optoacoustic tomography (MSOT) is a non-invasive imaging technique that illuminates tissue at many near-infrared (NIR) wavelengths and performs spectral detection of endogenous chromophores based on the absorption spectrum. The method has been shown capable to detect melanin, oxygenated and deoxygenated haemoglobin and lipids or externally administered photo-absorbing synthetic probes, within depths of several millimetres to centimetres[21–24]. A major challenge for implementing MSOT in basic research and clinical procedures is identifying moieties that enable contrast enhancement for improving detection of particular pathophysiological conditions. Melanin is found naturally in many living organisms and it absorbs strongly in the visible and NIR window[25]. Optimised as a natural absorber of light, melanin is therefore well suited for enhancing the contrast for optoacoustic imaging[26,27]. In addition, melanin has high photothermal conversion efficiency and is consequently highly suitable for photothermal therapy, i.e., treatment whereby light illumination of tissue that selectively contains melanin induces local heating that kills tumour cells[28]. The ability of melanin to serve both as a contrast enhancement and a therapeutic agent makes it appropriate for theranostics, i.e., the combination of diagnostics and therapeutics in a single agent. In that respect, melanin could not only provide contrast and improve the detection abilities for optoacoustic imaging, but also enable local therapy which can be monitored by optoacoustic imaging[29]. Harnessing the contrast enhancement and theranostic potential of melanin, however, requires overcoming its low solubility, which necessitates treatment with alkaline solvents and conjugation with hydrophilic polymers[27,28]. Such processing steps are costly and make scale up challenging. Therefore, developing a biological approach to package melanin inside cell membrane-derived nanocarriers would be of great benefit for diagnosis and therapy.

We hypothesise that we can package naturally occurring melanin into bacterial OMVs to create a biocompatible nanomaterial (OMV$^{Mel}$) for efficiently delivering the photoabsorber to target tissues for optoacoustic imaging and theranostic applications. In order to avoid systemic side effects due to bacterial endotoxin lipopolysaccharide (LPS), we use an *Escherichia coli* strain previously modified to be less endotoxic through inactivation of the *msbB* gene (to give OMV$^{ΔmsbB}$)[16], and we further engineer it to overexpress tyrosinase, which produces melanin that is passively incorporated into the cytosol and membrane of OMV$^{ΔmsbB}$ (to give OMV$^{Mel}$). We test the ability of MSOT to detect OMV$^{Mel}$ in phantoms and in vivo, and the vesicles generate strong MSOT signals in both cases. Systemically administered OMVs passively target and accumulate in tumour tissue via the enhanced permeability and retention (EPR) effect. We examine the ability of OMV$^{Mel}$ to produce local heating in vitro after irradiation with a pulsed NIR light source, and we confirm that this local heating retards tumour growth in vivo (photothermal treatment). Furthermore, we show that a single dose of OMV$^{Mel}$ inhibits tumour growth while triggering only mild, short-term systemic inflammation. These results establish OMV$^{Mel}$ as a promising agent for optoacoustic imaging and potentially theranostics, even suggesting that OMVs may be able to inhibit tumour growth through synergy of photothermal effects and cytokine-mediated antitumour responses[17]. In future, it may be possible to replace the melanin with other naturally derived theranostic cargos to generate a flexible platform for imaging-based theranostic applications against cancer and other diseases.

## Results

**OMV isolation, purification, and characterisation.** Figure 1 illustrates our approach to produce OMV$^{Mel}$. We started with a strain of *E. coli* K12 (derived from W3110) carrying an inactivated *msbB* gene, which leads to underacetylated and therefore much less endotoxic lipid A, an integral component of LPS[16,30,31]. We further engineered the bacteria to express *Rhizobium etli* tyrosinase[32], which is the rate-limiting enzyme in melanin biosynthesis. The idea was that melanin would be produced and would accumulate in the cytosol and periplasmic space, then it would be packaged into membrane and cytosol of OMVs that would be shed into the culture medium. OMVs were purified from bacteria carrying the *tyrosinase* gene (OMV$^{Mel}$), from the parental bacteria with an inactivated *msbB* gene (OMV$^{ΔmsbB}$), and from wild-type bacteria (OMV$^{WT}$) through multiple centrifugation steps. Figure 2a shows that the buffer containing isolated OMV$^{Mel}$ appeared black, whereas OMV$^{WT}$ appeared light brown. Figure 2b, c shows OMV characterisation in terms of size and morphology. As evident from hydrodynamic size and transmission electron microscope analysis, most vesicles were in the range of 20–100 nm and had a uniform circular, bilayer morphology. Their sub-micron size makes the OMVs suitable for in vivo optoacoustic imaging of tumours. The colour difference between wild-type and melanin-containing vesicles was evident in transmission electron micrographs. OMV$^{WT}$ was monitored after negative staining, whereas OMV$^{Mel}$ was visible without staining.

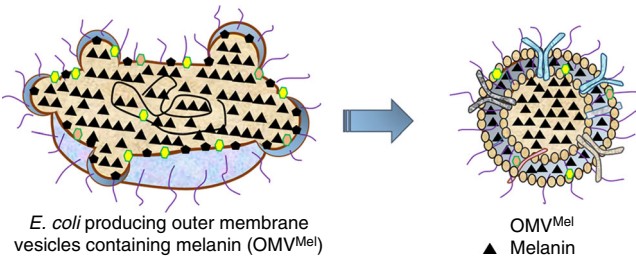

**Fig. 1** Schematic representation of OMV$^{Mel}$ generation. A schematic representation of OMV$^{Mel}$ purified after vesiculation from the parental bacteria. OMV, outer membrane vesicle

We tested whether the concentrated melanin inside OMV$^{Mel}$ could give a strong MSOT signal. Figure 2d shows that phantoms of OMV$^{Mel}$ gave ~7-fold stronger optoacoustic signal than phantoms of OMV$^{WT}$ after illumination at 750 nm. The absorption spectrum of OMV$^{Mel}$ resembled that of pure melanin[33], consistent with the idea that OMV$^{Mel}$ contains encapsulated melanin. These results suggest that OMV$^{Mel}$ can be used as an MSOT imaging probe, such as for basic studies of tumour biology, diagnostic studies and longitudinal monitoring of treatment response.

**Photothermal heating efficiency and cytotoxicity of OMV$^{Mel}$.** Melanin shows promise not only for disease research and diagnosis, but also for treatment. Irradiating the biopolymer can trigger local heating that kills diseased tissue through photothermal ablation or photothermal therapy[28,34]. We therefore asked whether the melanin packaged within OMV$^{Mel}$ shows potential to act as a nanoheater for photothermal therapy. Figure 3a shows that irradiating a suspension of OMV$^{Mel}$ led to much greater heating than irradiating OMV$^{WT}$ or phosphate-buffered saline (PBS), and that the rate of temperature increase was dependent on the concentration of melanin-carrying vesicles. Figure 3b shows that OMV$^{Mel}$ solutions increased by >10 °C from room temperature after only 2 min of laser irradiation, and that this heating effect persisted. This performance is promising, given preclinical studies showing that heating cancer cells to 42–50 °C for more than 5 min can be lethal[35–37]. Figure 3c shows that the experimentally determined absorption coefficient of OMV$^{Mel}$ was 0.75 cm$^{-1}$ (absorbance of 0.31 OD) at 750 nm. This is an order of magnitude greater than the absorption coefficient of tissue (approximately 0.1 cm$^{-1}$)[33], suggesting that photothermal effects can be induced selectively where OMV$^{Mel}$ deposits and not in neighbouring tissue. Figure 3d shows that OMV$^{Mel}$ exhibits photothermal conversion efficiency of 18.65% with a quantity of OMV$^{Mel}$ equivalent to approximately 150 µg of OMV$^{WT}$ (see Methods); this efficiency depends on melanin's concentration and molar absorptivity[1,2,38,39]. To test the potential for photothermal therapy in living cells, we exposed 4T1 breast cancer cultures to OMV$^{Mel}$, then irradiated the treated cells for several minutes with a tuneable nanosecond-pulsed laser operating from 730 nm to 830 nm. The total laser operating time was 6 min, which included both wavelength tuning and irradiation time. Figure 3e shows that the number of apoptotic cells, which appeared red because of their ability to bind ethidium homodimer-1, was much greater in cultures treated with OMV$^{Mel}$ than in cultures untreated or treated with OMV$^{WT}$.

**In vivo optoacoustic imaging of OMV$^{Mel}$.** As a first step towards validating the ability of OMV$^{Mel}$ to accumulate in tumours in vivo, we injected mice carrying subcutaneous 4T1 mouse mammary gland tumours with PBS, OMV$^{WT}$ or OMV$^{Mel}$ via the

tail vein. Figure 4a shows time-dependent accumulation of OMV$^{Mel}$ in tumour tissue using optoacoustic imaging. At 3 h after injection, unmixed melanin signal was much higher in animals treated with OMV$^{Mel}$ than in other animals. This strong optoacoustic signal was evident even at 24 h after injection, suggesting persistence in the circulation and, potentially, accumulation in tumours. Next, we defined regions of interest within tumours and quantified levels of melanin present. Figure 4b clearly indicates accumulation of OMV$^{Mel}$ in tumours via the EPR effect. Melanin signal was higher at 3 h than at 24 h after injection, indicating that some particles may undergo systemic clearance via the tumour vasculature. Given the overlap in absorption spectra between melanin and deoxyhaemoglobin, we wanted to confirm the presence of melanin based on spectral analysis. The spectral signature of deoxyhaemoglobin and melanin show lower absorption with increasing wavelength and are not orthogonal to each other. Thus, identification of melanin in tumours is a major challenge, in large part because tumours tends to be hypoxic relative to normal tissue, and hence deoxyhaemoglobin concentration is relatively high. Figure 4c, d compares the absorption spectra in the regions of interest in tumours from animals treated with PBS or OMV$^{Mel}$. As wavelength increased from 800 to 900 nm, optoacoustic signal decreased sharply (indicated with a black line) in the OMV$^{Mel}$ group but remained constant in the PBS group. These results are consistent with the idea that the spectra obtained from OMV$^{Mel}$-treated tumours are a combination of deoxyhaemoglobin and melanin spectra, whereas the spectra from PBS-treated controls reflect primarily deoxyhaemoglobin, for which the optoacoustic signal has been shown to remain constant between 800 and 900 nm[33]. In addition, the biodistribution of OMV$^{Mel}$ in a mouse was monitored during 120 min after intravenous injection (we did not examine non-tumour biodistribution over longer periods because we had to ensure that the animal did not move between measurements, and ethical considerations limited how long we could maintain the animal anaesthetised inside the MSOT set-up). During this 2 h period, the MSOT signal gradually increased in the tumour, tumour-adjacent region just below the skin as well as in liver and kidney (Supplementary Figure 1a). This provides further evidence that melanin is the primary source of contrast in our MSOT set-up, and it suggests that OMV$^{Mel}$ circulates and distributes in various organs. At the same time, the melanin signal in tumours rose consistently even from early time points (Supplementary Figure 1b) and our experiments at 3 and 24 h (Fig. 4a) showed appreciable signal in tumours. This persistent signal in the tumour suggests passive targeting ability of OMV$^{Mel}$, likely due to nanometre size that ensures penetration through the leaky blood vessels and due to EPR effects in the tumour region.

**In vivo photothermal therapy.** To assess the potential of OMV$^{Mel}$ for cancer treatment, we conducted photothermal experiments on nude mice bearing 4T1 tumours. Animals were randomly allocated to receive PBS intravenously or to receive OMV$^{WT}$ or OMV$^{Mel}$ intravenously or intratumourally. At 3 h after administration, animals were exposed (or not) to a continuous wave laser (1.5 W cm$^{-2}$, 800 nm) for 6 min. Tumour growth rate and animal body weight were recorded at fixed intervals. As can be seen in Fig. 5a, the tumour surface temperature in the laser-exposed area in animals treated intratumourally with OMV$^{Mel}$ reached as high as 56 °C, compared to 47 °C in animals treated intravenously with OMV$^{Mel}$ and 39 °C with PBS. This likely reflects much more efficient delivery of OMV$^{Mel}$ to tumours when they are injected directly. The much greater heating observed with intratumoural administration of OMV$^{Mel}$ was associated with superior therapeutic effect: in

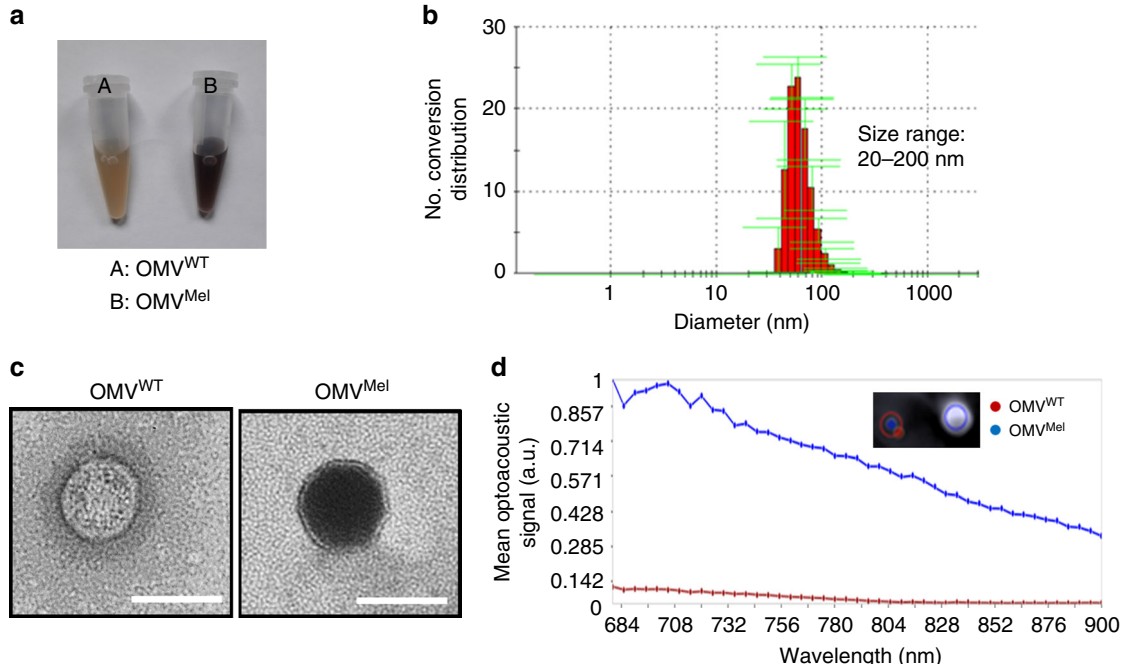

**Fig. 2** Outer membrane vesicle (OMV) purification and characterisation. **a** Purified form of OMV$^{WT}$ and OMV$^{Mel}$, isolated from parental bacteria by ultrafiltration and ultracentrifugation. **b** Dynamic light-scattering analysis of OMVs confirmed a particle size distribution in the range of 20 to 100 nm. **c** Transmission electron micrograph showing the nano-sized (<100 nm), bilayered, circular morphology of OMV$^{WT}$ and OMV$^{Mel}$. Scale bars, 100 nm. **d** Mean optoacoustic intensity (coloured line) as a function of wavelength for OMV$^{WT}$ and OMV$^{Mel}$

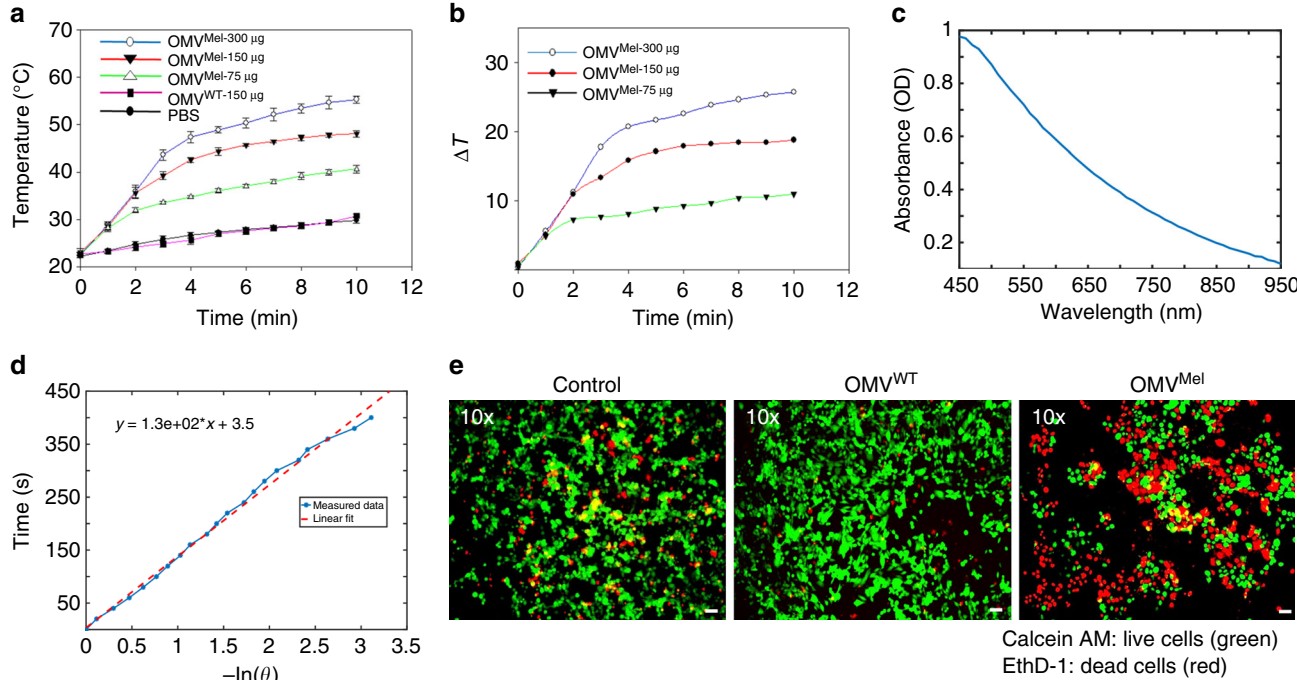

**Fig. 3** In vitro photothermal therapy using OMV$^{Mel}$. **a** Temperature curves of OMV$^{Mel}$, OMV$^{WT}$ and phosphate-buffered saline (PBS) during exposure to 750 nm light (650 mW cm$^{-2}$) over a period of 10 min. OMV, outer membrane vesicle. **b** Plot of temperature change ($\Delta T$) over a period of 10 min as a function of OMV$^{Mel}$ concentration. **c** Absorbance spectra of OMV$^{Mel}$ (~150 μg) solution obtained using a spectrometer. **d** Plot of time as a function of −ln ($\theta$) for the raw data and a linear fit during cooling after 10 min of irradiation as described for **a**. **e** Fluorescence images of 4T1 cells treated with PBS, OMV$^{WT}$ or OMV$^{Mel}$, then irradiated with a laser as described in the Methods. Viable cells were stained green with calcein-acetoxymethyl (AM), while dead cells were stained red with ethidium homodimer-1 (EthD-1). Scale bars, 20 μm

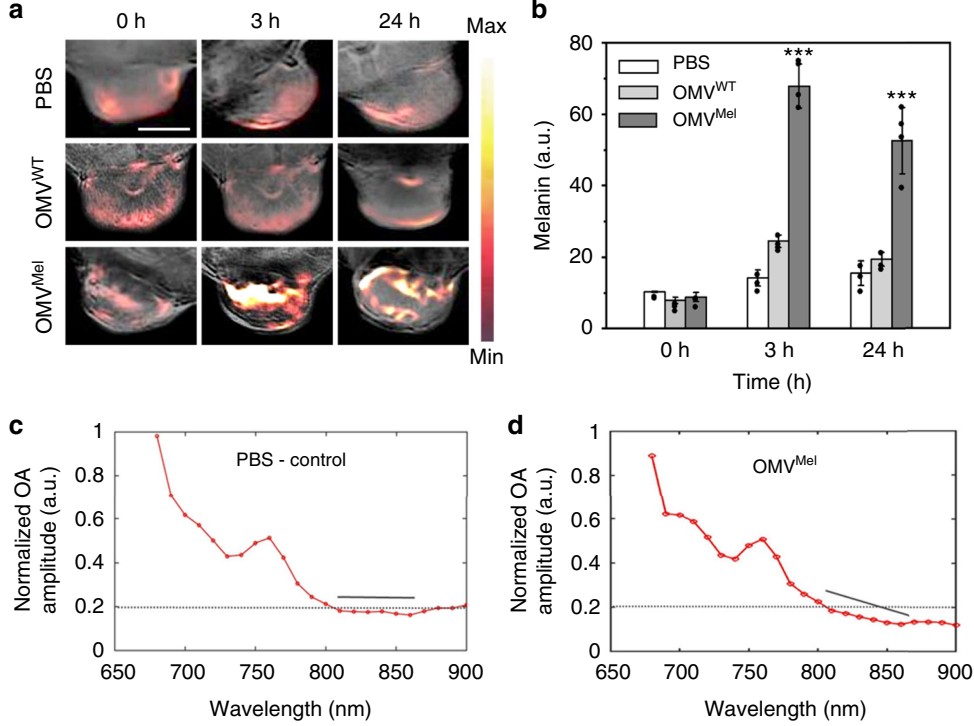

**Fig. 4** In vivo multi-spectral optoacoustic tomography (MSOT) imaging. **a** 4T1 tumour-bearing mice were given a single injection of phosphate-buffered saline (PBS) ($n=3$) or 150 μg of $OMV^{WT}$ ($n=4$) or $OMV^{Mel}$ ($n=4$) via the tail vein. Tumour-specific accumulation over time was monitored using a commercially available preclinical MSOT system. Scale bar, 4 mm. OMV, outer membrane vesicle. **b** Melanin concentration in the tumour was measured over time. A single mean value was calculated over the tumour region. Mean values and error bars were expressed as mean ± SD, inter-group differences were assessed for significance using the paired $t$-test compared to control and differences were considered significant if ***$p < 0.001$. **c**, **d** Optoacoustic spectra from the tumour region of animals treated with **c** PBS or **d** $OMV^{Mel}$

animals treated intratumourally and exposed to the laser, most tumour tissue appeared necrotic and the tumour mass nearly disappeared; in animals treated intravenously, tumour growth was reduced by approximately 43% as shown in Fig. 5b. No antitumour effects were observed in animals treated with PBS or in animals treated with OMVs in the absence of laser exposure. Throughout these phototherapy experiments, animals in all groups appeared normal and showed stable body weight (Fig. 5c), with no overt signs of toxicity. Our results suggest that $OMV^{Mel}$ can be used for in vivo optoacoustic imaging and phototherapy.

**In vivo safety and immune responses**. To confirm the in vivo safety of OMVs for photothermal therapy, we investigated whether single-dose systemic injection of PBS or OMVs ($OMV^{WT}$, $OMV^{\Delta msbB}$ or $OMV^{Mel}$) would stimulate the immune system in C57BL/6 mice. The dose of OMVs was 75 μg, similar to the photothermal therapy experiment. Then, levels of the cytokines tumour necrosis factor-α (TNF-α), interleukin-6 (IL-6) and interferon-γ (IFN-γ) in serum were measured by enzyme-linked immunosorbent assay (ELISA). Cytokine levels were evaluated at 2 and 24 h in order to monitor early and delayed immune responses[16]. All three types of OMVs increased serum levels of the three cytokines at 2 h, with $OMV^{WT}$ triggering the greatest increases (Supplementary Figure 2a). In all cases, cytokine levels decreased close to baseline by 24 h. In addition, histology of heart, liver, spleen and kidney at 24 h after injection did not indicate significant organ damage under these treatment conditions (Supplementary Figure 2b). No animal mortality occurred in any of the groups. These results suggest that the underacylated LPS on $OMV^{\Delta msbB}$ and $OMV^{Mel}$ induce milder systemic inflammation than the intact LPS on $OMV^{WT}$, and that the modified OMVs are well tolerated upon systemic administration.

## Discussion

Here we describe melanin-containing OMVs as a biocompatible contrast agent that may allow longitudinal imaging of tumours in vivo using optoacoustics, which offers several advantages over fluorescence methods for cancer monitoring. Fluorescence imaging does not offer good spatial resolution deeper than 0.5 mm because tissue strongly scatters light[40–43]. Optoacoustics, in contrast, generates optical images by recording the acoustic waves, which tissue does not strongly scatter[21,44]. In addition to offering images of higher resolution from deeper-lying tissue, optoacoustics is better suited than fluorescence for theranostics. Fluorescent agents cannot optimally emit fluorescence and heat target tissue locally, because the two processes of fluorescence and nonradioactive decay compete with each other. In the case of optoacoustics, contrast agents can be optimised for photothermal conversion efficiency, such that they simultaneously give strong acoustic signal while also converting light energy to heat.

We showed with transmission electron microscopy that naturally produced OMVs was able to effectively package melanin expressed from a transgene. Melanin was presumably encapsulated into OMVs together with various cytosolic-, periplasmic- or membrane-bound components during vesicle generation and secretion, though the exact mechanism for melanin packaging is not known[45,46]. Importantly, the generated $OMV^{Mel}$ particles remained sub-micrometre in size, with intact circular morphology, which indeed is similar to previously reported bioengineered OMVs under study for cancer-specific drug delivery[16]. $OMV^{Mel}$ show good dispersion properties and small particle size, allowing them to remain in circulation for an extended time, which facilitates passive accumulation in the tumour tissue by the EPR effect. We performed these experiments in a bacterial strain with a mutated form of the *msbB* gene. The resulting under-acylation

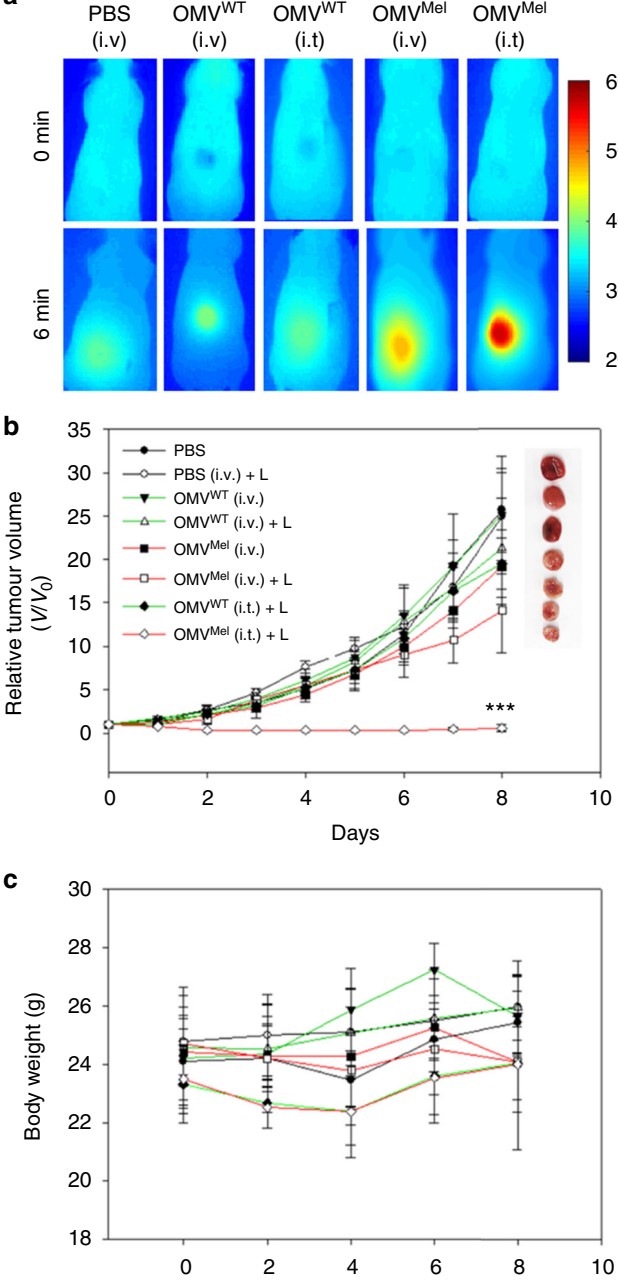

**Fig. 5** In vivo photothermal therapy. **a** Infrared (IR) thermal images of 4T1 tumour-bearing mice before and after laser irradiation (1.5 W cm$^{-2}$, 800 nm, 6 min). Before irradiation, animals were injected with phosphate-buffered saline (PBS) intravenously or with OMV$^{Mel}$ or OMV$^{WT}$ intravenously (i.v.) or intratumourally (i.t.). **b** Tumour growth curves. Representative images of dissected tumours are also shown, except for OMV$^{Mel}$ (i.t.) + L, which had nearly disappeared. Mean values and error bars are presented as mean ± SD, inter-group differences were assessed for significance using the paired $t$-test compared to control (***$p < 0.001$ vs. PBS with laser treatment; $n = 4$). **c** Body weight of all animals was recorded during each treatment, with all animals appearing healthy throughout the study based on eating and behaviour. OMV, outer membrane vesicle; L, laser irradiation

of lipid A substantially reduces the endotoxic activity of lipopolysaccharide, leading to weaker Toll-like receptor-dependent host immune reactions and increased in vivo safety[16,17,30,47]. The same strain has previously been used to develop OMVs for in vivo

drug delivery applications[16]. Using less endotoxic bacterial strains is attractive for facilitating translational research on bacterially derived particles[16,17]. Furthermore, the produced melanin was non-toxic to the bacterial cells. The entire process of melanin encapsulation is a natural event, so it does not require any sophisticated synthesis skills, and OMV$^{Mel}$ can easily be scaled up cost-effectively using well-established large-volume bacterial culture and purification techniques.

The absorption coefficient of OMV$^{Mel}$ was found to be severalfold higher than that of tissue. As a result, OMV$^{Mel}$ is expected to show strong optoacoustic signal. It appears that the proteolipid envelope of OMVs does not influence the absorption properties of encapsulated melanin biopolymers. In addition to NIR absorption, OMV$^{Mel}$ show suitable properties for deep-tissue optoacoustic imaging and photothermal therapy. It offers a photothermal conversion efficiency of 18.6%, and it allows local heating in a time- and melanin-dependent manner. Within 2 min of NIR laser irradiation, the temperature of OMV$^{Mel}$ suspensions increased beyond 45 °C, which is sufficient to kill cancer cells in vivo[35–37,48]. Since body temperature is around 37 °C, these results suggest that OMV$^{Mel}$ that has accumulated into tumours can easily be heated above 45 °C upon NIR irradiation. When we irradiated OMV$^{Mel}$-treated 4T1 cancer cells in culture, massive cytotoxic effects were observed that were consistent with good photothermal conversion efficiency. Next, we demonstrated the efficacy of photothermal therapy in vivo. Maximal temperature on the tumour surface reached 56 and 47 °C after intratumoural or intravenous injection of OMV$^{Mel}$, compared to 39 °C in PBS-treated controls. Regardless of whether OMV$^{Mel}$ was delivered intratumourally or intravenously, it triggered obvious necrosis after laser exposure, and the tumour was nearly completely destroyed in animals treated intratumourally. These findings demonstrate the potential of OMV$^{Mel}$ as an effective nanoheater for diagnostic imaging and thermal therapy. Our in vivo photothermal therapy experiments were performed with continuous wave laser illumination. Our group previously demonstrated that photodynamic therapy can be performed with pulsed laser illumination[29], and hence it should be possible to heat tumours with a nanosecond-pulsed laser illumination of MSOT as long as sufficient power can be delivered. In this case, MSOT could be used for image-guided therapy. Optoacoustic measurements can be used to estimate temperature distribution during photothermal therapy[49], and hence an ideal theranostic system may combine phototherapy based on OMV$^{Mel}$ with continuous monitoring of local temperature and OMV$^{Mel}$ distribution using MSOT.

Due to its natural origin, OMV$^{Mel}$ is a biocompatible and biodegradable agent that can be produced on a large scale in a cost-effective manner[16,50–52]. Here we explored its potential as a biologically derived theranostic agent. Our work has shown noninvasive monitoring of OMV$^{Mel}$ using MSOT, and we demonstrated light-based heating with OMV$^{Mel}$ in vitro and in tumour-bearing athymic nude mice. We used nude mice because their lack of pigmentation and skin hair facilitate optoacoustic analysis. However, this prevented us from determining whether OMV$^{Mel}$ can stimulate immune responses against tumours. OMVs have been shown to induce the production of antitumour cytokines, including IFN-γ, in tumours of immunocompetent mice[17]. Repeated administration of OMVs can induce sustained production of IFN-γ in tumour tissue as well as long-term immune memory[17]. Further detailed work is needed to examine whether, as suggested for other OMVs[17], our OMV$^{Mel}$ can synergistically attack tumours by locally heating the tissue and triggering a longterm immune response to prevent cancer recurrence; indeed, hyperthermia can increase blood flow and immune cell infiltration.

OMVs carry several proteins and LPS on membrane surface, which might trigger systemic toxicity[16–20]. In order to address safety issues of OMVs, we carried out a separate study using immunocompetent mice. OMV[WT] induced substantially stronger Toll-like receptor responses than OMV[ΔmsbB] and OMV[Mel] based on serum cytokine assays (Supplementary Figure 2a), reflecting that the latter two OMV types were prepared from a low-endotoxicity *E. coli* that is generally well tolerated. The OMV platform appears less likely to induce chronic systemic toxicity or other side effects, based on reports of bioengineered OMVs that were repeatedly injected into a mouse cancer model[16,17]. An important question is what happens to OMV[Mel] or melanin, which can function as a self-antigen. Studies with other OMVs suggest that they are degraded in late endosomes or lysosomes[16,50,51], and that intracellular melanin is broken down by NADPH-dependent oxidoreductase[53]. The melanin in intact OMV[Mel] may remain hidden from the immune system because it is embedded within the bacterial membrane. Future work should verify and extend our findings by checking for systemic toxicity or toxic effects in major organs upon repeated administration of modified OMVs in small and large animals. Also, potential synergies between photothermal therapy mediated by melanin and immune responses triggered by the OMVs themselves should be explored. Long-term stability of OMV[Mel] upon standing should be evaluated, since this is important for clinical applications.

The efficacy of OMV[Mel] as a specific anti-cancer therapy depends on its ability to target tumours, and here we demonstrate such an ability in vivo. Optoacoustic signal due to OMV[Mel] was evident in tumours soon after intravenous injection, and this signal increased during 120 min, suggesting tumour accumulation via the EPR effect. At the same time, appreciable melanin signal from OMV[Mel] was observed in a tumour-adjacent region just below the skin as well as in liver and kidney. This suggests that OMV[Mel] can persist in other organs, and the implications of this for safety should be explored further. It may not necessarily have toxic effects, since photothermal therapy usually involves shining activating light only on the tumour. The present work focused on imaging biodistribution of OMV[Mel] using MSOT. Future work should examine the fate of intact OMV[Mel] and the released melanin in vivo in greater detail.

Contrast enhancement in optoacoustic imaging originally relied on metal particles, in particular gold nanoparticles[54,55]. Although such particles can be produced with desired physical dimensions (size and shape) in a highly controlled way, they present several disadvantages for human use. They are difficult to reproduce on large scale and are vulnerable to photobleaching, which gives rise to variable optoacoustic response, and they can persist for long periods in tissue, increasing the risk of toxic effects[54,55]. We and others have shown that the use of dye-based nanoparticles absorbing in the NIR, such as quenched hexacene dye nanoparticles (referred as QH$_2$ particles), can be a potent alternative to NIR metal nanoparticles for strong optoacoustic signal generation[56]. Importantly, QH$_2$ particles can quench unwanted fluorescence and offer greater photostability in the NIR region than metal compounds. We also developed polymethine dyes such as DY-635 and DY-780 conjugated to polymeric scaffolds, and we showed that they could be used for organ-selective elimination, optoacoustic imaging and drug delivery; in fact, DY-780 proved to be suitable for MSOT imaging[57]. The work herein takes the next step in generating bioengineered optoacoustic probes with the potential to seamlessly translate into clinical and commercial successes. Importantly, advances in biosynthesis and bioprocessing have provided scalable and robust platforms for manufacturing of such bioengineered theranostics to meet clinical needs.

MSOT operates in the NIR spectral region, where penetration of tissue to depths of several centimetres becomes possible because of low light attenuation, and laser exposure of the sample remains within the ANSI (American National Standards Institute) biosafety limits[58]. High molecular detection specificity is achieved by resolving (unmixing) multiple spectral signatures in tissues[22]. Clinical translation of MSOT is limited due to the lack of imaging probes that are biocompatible and scalable. Bioengineered OMVs or similar cell membrane-derived particles exhibit improved theranostic efficacy as well as reduced toxicity[16,17,19]. Preclinical success of such biological nanoparticles for diagnostic or therapeutic applications can have paradigm-shifting effects on clinical or commercial translation of biological agents.

Here we acquired MSOT data with a transducer array having 256 channels covering a 270° field of view (in a two-dimensional plane). Light was delivered using a fibre bundle with six outputs, but high photothermal efficiency requires delivering light at a specific location of the body with a single output fibre. This can be achieved using handheld systems with single-point illumination or line illumination, but such systems feature limited detection angles, i.e., 145° in a two-dimensional set-up, and 90° in a three-dimensional set-up. As a result, quantification is challenging. Accurate quantification methods are being designed based on sparse recovery[59], which may enable real-time three-dimensional theranostic applications of OMV[Mel].

We report a bacterial membrane-derived nanoparticle system for optoacoustic imaging. The entirely biological design of this system may effectively avoid the long-term toxicity issues associated with many synthetic agents that deposit in various tissues and exhibit poor clearance. The ability to produce OMV[Mel] from bacteria means that the vesicles can be loaded with diverse genetically encoded agents suitable for optoacoustic imaging and image-guided therapy[60]. Potentially useful cargo proteins include fluorescent proteins, several optoacoustic-suitable chromoproteins and phytochromes[61,62], anti-cancer toxins[63] and peptides (e.g., aptides)[64] or proteins (e.g., affibodies)[14,16,65] that target specific tissues and therefore provide stronger therapeutic efficacy with weaker side effects. We anticipate that the OMVs presented in this work will be a starting point for future theranostic studies focused on simultaneous optoacoustic imaging and cancer therapeutics.

## Methods

**Plasmid construction, bacterial strain and growth.** The *melA* gene of *Rhizobium etli* encoding tyrosinase (kindly provided by Professor Guillermo Gosset, Universidad Nacional Autónoma de México, Mexico)[32] was cloned into pGEX-4T-1 (GE Healthcare, Freiburg, Germany). The construct pGEX-4T-1-melA was transformed into msbB mutant W3110-K12 *E. coli* (kindly provided by Professor Sangyong Jon, KAIST, South Korea)[16]. Bacteria were cultured in 1.5-L flasks at 30 °C and 180 rpm for 5 days. Tyrosinase production was induced at OD 0.6 using isopropyl β-ᴅ-1-thiogalactopyranoside (IPTG) at a final concentration of 0.5 mM. To support melanin production the media were supplemented with 94.5 mg CuSO$_4$ as well as 1 g ʟ-tyrosine. OMV[Mel] were produced in msbB mutant W3110-K12 *E. coli* transformed with pGEX-4T-1-melA plasmid. OMV[WT] were produced from untransformed msbB mutant W3110-K12 *E. coli*.

**OMV purification and characterisation.** Based on our previous work[16], OMVs were produced as follows. *E. coli* were cultured as described above. Next, bacterial cells were removed by centrifugation at 7500 × *g* for 45 min at 4 °C. The resulting supernatant was filtered by passing through a 0.45-μm membrane filter (Nalgene, Thermo Scientific), and concentrated to 100 mL using 100-K ultrafiltration membrane. The concentrate was further precipitated using ammonium sulphate (at final concentration, 400 g L$^{-1}$) at 4 °C overnight. Crude OMVs were obtained by centrifugation at 12,000 × *g* for 45 min, the resulting pellet was resuspended in 1 mL PBS and was layered over a sucrose gradient (1 mL each of 2.5, 1.6 and 0.6 M sucrose) and separated from free melanin by ultracentrifugation at 150,000 × *g* for 3–5 h at 4 °C. The collected OMV fractions were washed with PBS with centrifugation at 150,000 × *g* for 1–2 h at 4 °C, resuspended in 1 mL PBS containing 15% glycerol, filtered through 0.45-μm cellulose acetate filters and stored at −20 °C until use. In the case of OMV[WT], total protein concentration was estimated using

the bicinchoninic acid (BCA) assay (Thermo Scientific), and this was defined to be the OMV$^{WT}$ concentration. Since black colour prevents protein estimation with the BCA assay, we had to develop an alternative procedure to determine OMV$^{Mel}$ concentration. Concentrations of OMV$^{Mel}$ solutions were estimated based on the size of the pellet relative to the pellet obtained from OMV$^{WT}$ solutions (by ultracentrifugation) of known concentration. We verified the accuracy of this approach by comparing intensities of major protein bands obtained after fractionating OMV$^{WT}$ and OMV$^{Mel}$ on sodium dodecyl sulphate–polyacrylamide gel electrophoresis. OMV samples were characterised with respect to size and morphology using an electrophoretic light-scattering apparatus (Malvern Zetasizer) and transmission electron microscope (Zeiss Libra 120 Plus, Carl Zeiss NTS GmbH, Oberkochen, Germany).

**Cell culture and in vivo tumour experiments.** 4T1 mammary gland carcinoma (CRL-2539, ATCC, Manassas, VA, USA) was cultured in RPMI-1640 medium supplemented with 10% foetal bovine serum and antibiotics. Cells were incubated at 37 °C in 5% CO$_2$. Animal procedures were reviewed and approved by the Animal Care and Handling Office of Helmholtz Zentrum München and by the Government of Upper Bavaria. Sample sizes were chosen based on guidance from the literature. Investigators were not blinded to the identity of groups. Female athymic Fox-N-1 nude mice 6 weeks old (Envigo, Germany) were implanted subcutaneously with 4T1 murine breast cancer cells ($0.8 \times 10^6$ cells per animal). Tumour volume, calculated as (width)$^2 \times$ (length)$\times 1/2$, and relative tumour volume, calculated as $V/V_0$ (where $V_0$ was the tumour volume when the treatment was initiated), was monitored at regular intervals and was expressed as group mean ± standard deviation (SD). Tumour growth inhibition (TGI) was determined on the final day as % TGI: $100\% \times \left(T_{vol}^{PBS} - T_{vol}^{OMV}\right) \times \left(T_{vol}^{PBS}\right)^{-1}$, where $T_{vol}$ is final tumour volume − initial tumour volume.

**MSOT set-up and data acquisition.** Phantom and mice data were acquired using a commercially available MSOT scanner (MSOT256-TF, iThera Medical GmbH, Munich, Germany)[66]. Nanosecond-pulsed light was generated from a tuneable optical parametric oscillator (OPO) laser and delivered to the sample through a ring-type fibre bundle. The wavelength used for imaging was from 680 to 900 nm with a step size of 10 nm and 10 averages. Light is absorbed by the sample and generates an acoustic signal that propagates through the sample and is detected outside the sample. In our experiments, acoustic signals were detected as time-series pressure readouts at 2030 discrete time points at 40 Mega samples per second using a cylindrically focused, 256-element transducer. The transducer array had a central frequency of 5 MHz (−6 dB was approximately 90%) with a radius of curvature of 40 mm and an angular coverage of 270°.

Acoustic data were measured using a transducer array with 256 elements and 270° coverage at multiple laser wavelengths. The acquired acoustic data were filtered using a Chebyshev filter with cut-off frequencies of 0.1–7 MHz. Optoacoustic images were reconstructed using a Tikhonov regularisation-based scheme using filtered optoacoustic data; the regularisation parameter was chosen automatically using L-curve methods[67]. The Tikhonov method was implemented using the least-squares QR method to reconstruct an image at each wavelength[59]. These spectral optoacoustic images were used to unmix melanin chromophores through a linear unmixing method that assumed that fluence did not influence unmixing. Melanin quantification was then performed on the unmixed melanin image from the tumour region by plotting mean intensity from a defined region of interest (corresponding to a similar tissue area in all animals) over time. Differences in mean melanin levels were considered significant if $p < 0.01$. Advanced unmixing methods are being developed to quantify biomolecules[68], but these methods have been extensively validated only for oxy- and deoxyhaemoglobin. Efforts are underway to extend these methods to other biomolecules such as melanin and lipids.

**MSOT imaging of phantoms.** A cylindrical agar phantom[69] was prepared with two cylindrical holes, one for OMV$^{WT}$ and another for OMV$^{Mel}$. The cylindrical phantom contained 1.3% (w/w) agar (Sigma-Aldrich, St. Louis, MO, USA) to provide solidity and 6% (v/v) intralipid emulsion (20%, Sigma-Aldrich) for light diffusion to enable uniform illumination of the holes. The added intralipid gives a reduced scattering coefficient of 10 cm$^{-1}$, mimicking scattering in tissue[33]. The two cylindrical holes were 3 mm in diameter. MSOT data were acquired as described above to allow transversal plane imaging at a single position, located approximately in the middle of the phantom.

**Measurement of photothermal activity.** Photothermal activity was measured by placing PBS or OMV$^{WT}$ (150 μg mL$^{-1}$) or OMV$^{Mel}$ solution (75, 150 or 300 μg mL$^{-1}$ equivalent concentration) in a quartz cuvette, and then exposing solutions to an NIR light source at 750 nm (Tunable Optical Parametric Oscillator Laser, InnoLas Laser, Krailling, Germany) while monitoring temperature changes with a digital thermometer for up to 10 min. Concentration-response experiments were performed at a fixed laser power (650 mW cm$^{-2}$) and compared with PBS as a control.

**Cytotoxicity study.** Fluorescence-based live/dead cell assays were carried out using the molecular probes calcein-acetoxymethyl (AM) and ethidium homodimer-1 (Invitrogen). Briefly, cells were seeded into 96-well plates and incubated with or without OMV$^{WT}$ or OMV$^{Mel}$ for 6 h. After the media were replaced with fresh media, cells were exposed to nanosecond-pulsed laser treatment (730–830 nm, 650 mW cm$^{-2}$) for 6 min and then stained with calcein-AM and ethidium homodimer-1 following the manufacturer's instructions. Cells were examined under a fluorescence microscope to determine numbers of live and dead cells (Thermo Scientific).

**MSOT imaging of mice and estimation of melanin.** Animal procedures were reviewed and approved by the Animal Care and Handling Office of Helmholtz Zentrum München and by the Government of Upper Bavaria. To examine in vivo cancer targeting, mice bearing 4T1 tumours (~150 mm$^3$) were given a single injection of either PBS, OMV$^{WT}$ (~150 μg) or an equivalent amount of OMV$^{Mel}$ via the tail vein ($n = 4$ for OMV treatment and $n = 3$ for PBS treatment). An additional mouse was scanned to monitor distribution of OMV$^{Mel}$ in tumour, liver and kidney from 0 to 120 min after intravenous injection of OMV$^{Mel}$ (mouse was killed immediately after imaging). Mice were anaesthetised with 1.8% isoflurane in oxygen, which was delivered throughout MSOT data acquisition. Ultrasound gel was applied to the mice before acquisition to enable coupling between the tissue and water medium. Tissue homing and retention were monitored at 3 h and 24 h post injection with MSOT. In a separate experiment, OMV$^{Mel}$ (~75 μg) was injected via the tail vein, and then the tumour and nearby region (just below the skin) were monitored with MSOT continuously for 120 min without moving the mouse (mouse was killed immediately after imaging). Melanin intensities are averages calculated from 6 adjacent slices from the tumour centre. Sample sizes for animal studies were chosen based on institutional recommendations with guidance from the literature. Investigators were not blinded to animal group allocations.

**In vivo photothermal therapy.** Tumour volume and body weight were recorded at regular intervals. Once the tumour volume reached approximately 100 mm$^3$, mice were divided into eight groups ($n = 4$ mice/group). The groups were treated with PBS intravenously, with or without laser treatment; with OMV$^{WT}$ or OMV$^{Mel}$ (each ~75 μg) intravenously, with or without laser treatment; and with OMV$^{WT}$ or OMV$^{Mel}$ intratumourally, with laser treatment. At 3 h after injection, tumours were irradiated with a continuous wave laser (1.5 W cm$^{-2}$, 800 nm) for 6 min. Temperature at tumour sites was recorded immediately before and after laser irradiation using an IR thermal camera (FLIR i60). Investigators were not blinded to the identity of groups.

**In vivo safety and immune responses.** The safety and immune responses to OMVs were assessed in female C57BL/6 mice 6 weeks old. Animals were divided into four groups ($n = 5$ mice/group): (1) vehicle control (PBS), (2) OMV$^{WT}$, (3) OMV$^{\Delta msbB}$ and (4) OMV$^{Mel}$. OMVs were administered intravenously, then blood was collected via cardiac puncture (under surgical anaesthesia) at 2 and 24 h. Levels of the TNF-α, IL-6 and IFN-γ in serum were quantified at 450 nm using commercial ELISA kits (R&D Systems) according to the manufacturer's instructions. Vital organs including heart, liver, spleen and kidney were extracted, fixed and frozen in OCT embedding gel. Tissue blocks were sectioned to 10 μm, and stained with haematoxylin and eosin. Images were obtained by light microscopy (Carl Zeiss). Investigators were not blinded to the identity of the groups.

**Statistics.** Sample sizes were chosen based on guidance from the literature. Animals of the same gender, age and genetic background were randomised for grouping. Statistical analyses were performed using SPSS 18.0 (IBM, Chicago, IL, USA). Inter-group differences were assessed for significance using the paired $t$-test. Results were expressed as mean ± SD, and differences were considered significant if $p < 0.01$.

## Data availability

All data presented in the paper are available from the authors upon reasonable request.

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

## Acknowledgements

This project has received funding from the European Research Council (ERC) under the European Union's Horizon 2020 research and innovation programme under grant agreement no. 694968 (PREMSOT). The research leading to these results was supported by the Deutsche Forschungsgemeinschaft (DFG), Germany (Gottfried Wilhelm Leibniz Prize 2013, NT 3/10–1) as well as by the DFG as part of the CRC 1123 (Z1). J.P. acknowledges support from the Alexander von Humboldt Postdoctoral Fellowship Program. We wish to thank Professor Guillermo Gosset (Universidad Nacional Autónoma de México) for providing the plasmid encoding *R. etli* tyrosinase, Professor Sangyong Jon (Korea Advanced Institute of Science and Technology, South Korea) for providing msbB mutant W3110-K12 *E. coli*, Andreas Schroeppel and Dr. Otmar Schmid (Comprehensive Pneumology Center Munich, Institute of Lung Biology and Disease, Helmholtz Zentrum München) for helping us perform particle size measurements in their laboratory, Dr. Juan Antonio Aguilar-Pimentel (German Mouse Clinic/Institute of Experimental Genetics, Helmholtz Zentrum München) for providing the IR thermal camera, and Kanuj Mishra, Nian Liu, Dr. Doris Bengel, Ruth Hillermann, Sarah Glasl and Pia Anzenhofer for assisting with experimental procedures. We also wish to thank Dr. A. Chapin Rodríguez for helpful suggestions on the manuscript.

## Author contributions

V.G. conceived the OMV^Mel system and designed the study. V.G. and J.P. performed the experiments, J.P. processed the MSOT data and helped analyse the data. J.M.-N. assisted with in vivo experiments and MSOT data analysis. A.S. subcloned the *melA* gene and assisted with bacterial culture. U.K. assisted with in vivo experiments and animal maintenance. G.M. and V.G. performed transmission electron microscopy, which M.A. and A.W. supervised. V.N. provided significant intellectual input, helped interpret the results and supervised the research. All authors contributed to writing the paper.

## Additional information

**Competing interests:** V.N. is a shareholder in iThera Medical GmbH, Munich, Germany. The remaining authors declare no competing interests.

