## [Peer Review File · Nature Communications]

Reviewers' comments:

Reviewer #1 (Remarks to the Author):

This manuscript, "Bioengineered bacterial vesicles: a novel class of biological nano-heaters for optoacoustic imaging", developed bacterial outer membrane vesicles (OMVs) encapsulating biopolymer-melanin for tumor targeting and photoacoustic imaging contrast enhancement. The authors developed nanocarriers using OMVs, which have higher biocompatibility and stability than nanomaterials using organic, inorganic materials and metals, and increased optical absorbance using melanin. The developed materials can be easily synthesized naturally, unlike conventional nanomaterials, which have complex synthetic methods. The developed materials showed good optical properties for photoacoustic imaging. At the end, the tumor targeting efficiency was successfully monitored with photoacoustic imaging. After addressing the following comments, this work can be publishable in Nature Communication.

Comments

1. The concept of OMVs for tumor targeting and therapy has been developed in other studies such as [Kim, Oh Youn, et al. "Bacterial outer membrane vesicles suppress tumor by interferon- γ -mediated antitumor response." Nature communications 8.1 (2017): 626.], and [Gujrati, Vipul, et al. "Bioengineered bacterial outer membrane vesicles as cell-specific drug-delivery vehicles for cancer therapy." ACS nano 8.2 (2014): 1525-1537.] and the tumor targeting process was monitored with fluorescence imaging. What is the novelty of OMVMel –compared to other studies and what are the benefits of monitoring the cancer targeting efficiency of OMVs using optoacoustic imaging rather than fluorescence imaging?
2. The bacterial outer membrane vesicles (OMVs) can cause immune response in biological environment and the OMVs are less toxic than the bacteria itself but may be toxic. Did the developed OMVMel show no immune response? If not, did you make any special treatment to stop the immune response?
3. How is OMVMel released from the body after it has accumulated in the cancer? Does OMVMel accumulate in other organs, such as the liver or kidneys? It would be nice if the authors could add photoacoustic images of other organs after OMVMel injection.
4. In Measurement of photothermal activity section, there is a typo. "Tunable optical parametric oscillator laser \diamond Tunable optical parametric oscillator laser"
5. A scale bar should be added in Figure 4. (a).
6. Make Y-axis scale of figure 4. (c) and (d) the same for a better comparison.

Reviewer #2 (Remarks to the Author):

This study demonstrates the use of bioengineered vesicles as a potent agent for optoacoustic imaging, with the potential to enable both image enhancement and hyperthermia applications. This manuscript was well prepared and the results were also described well. However, there is a pity in the contents of the study results. This paper emphasizes the use of bioengineered vesicles containing melanin as a theranostic agent. Therefore, the authors should propose results for cancer therapy in vivo by hyperthermia. In addition, the experimental method needs to be presented in more detail. For instance, the method for the measurement of melanin amount in the tumor tissue should be described. Unfortunately, I do not recommend publishing this manuscript.

Reviewer #3 (Remarks to the Author):

This study describes an interesting and novel application for bacterial outer membrane vesicles: delivery of melanin chromophore for optoacoustic imaging and therapy applications. An ingenious method for OMV incorporation is used, and the described results provide a first proof of principle for this approach.

I have the following questions about the results:

1. Figure 3 shows temperature curves for OMVs(Mel) but not for the empty OMVs. It would be helpful to include this control as well. Similarly, panel 3e shows the cytotoxic effect after irradiation for the OMVs(Mel) but lacks the control without melanin.

2. If I understand Fig.4 correctly, it only shows data for tumor tissue. A comparison with other tissues is necessary to validate the targeting specificity.

More general remarks:

1. I am not familiar with the term "theranostics". Please clarify.

2. Melanin is a self-antigen. It is possible that presentation through OMVs will lead to immunogenicity of melanin, as OMVs are known to be an efficient vaccine delivery system. So there is a theoretical risk of breaking tolerance and inducing autoimmunity. This may become a problem when clinical application is considered. Please discuss this issue.

Reviewers' comments:

Reviewer #1 (Remarks to the Author):

This manuscript, "Bioengineered bacterial vesicles: a novel class of biological nano-heaters for optoacoustic imaging", developed bacterial outer membrane vesicles (OMVs) encapsulating biopolymer-melanin for tumor targeting and photoacoustic imaging contrast enhancement. The authors developed nanocarriers using OMVs, which have higher biocompatibility and stability than nanomaterials using organic, inorganic materials and metals, and increased optical absorbance using melanin. The developed materials can be easily synthesized naturally, unlike conventional nanomaterials, which have complex synthetic methods. The developed materials showed good optical properties for photoacoustic imaging. At the end, the tumor targeting efficiency was successfully monitored with photoacoustic imaging. After addressing the following comments, this work can be publishable in Nature Communication.

Response: We thank the reviewer for the insightful discussion and suggestions. We have addressed all comments and concerns raised as discussed in the following:

1. The concept of OMVs for tumor targeting and therapy has been developed in other studies such as [Kim, Oh Youn, et al. "Bacterial outer membrane vesicles suppress tumor by interferon- γ -mediated antitumor response." Nature communications 8.1 (2017): 626.], and [Gujrati, Vipul, et al. "Bioengineered bacterial outer membrane vesicles as cell-specific drug-delivery vehicles for cancer therapy." ACS nano 8.2 (2014): 1525-1537.] and the tumor targeting process was monitored with fluorescence imaging. What is the novelty of OMVMel compared to other studies and what are the benefits of monitoring the cancer targeting efficiency of OMVs using optoacoustic imaging rather than fluorescence imaging?

Response 1: As suggested we have edited the manuscript to highlight the advances of developing OMV^{Mel} for optoacoustic imaging in contrast to fluorescence imaging. Please see revised text in Page 11 and 12 of revised manuscript.

Briefly, in the current study we report OMV^{Mel} specifically prepared for optoacoustic imaging (OAI) and photothermal therapy (PTT) applications, i.e. in relation to enhancing optoacoustic contrast and considering the vesicles for theranostic and therapeutic (photothermal) applications. The previous two reports were instead focused on drug-delivery and immunotherapy applications of OMVs.

Moreover, fluorescence imaging is a two-dimensional imaging technique which comes with great limitations in resolving depth or achieving high resolution and accuracy when considering macroscopic planar imaging of tissues. The method suffers from poor spatial resolution due to photon scattering in tissues, which challenges imaging performance. Therefore accurate localization and imaging of fluorescent probe labelled OMVs or other nanoparticles in-vivo is not possible at depths deeper than ~ 0.5 mm, relying only on fluorescence. Conversely, optoacoustic imaging detects optical contrast (optical absorption, not fluorescence) based on acoustic (ultrasound) signals and achieves high-resolution three-dimensional optical imaging in tissues. Therefore it offers unprecedented performance compared to fluorescence imaging, since acoustic scattering is several orders of magnitude weaker than optical scattering. Therefore optoacoustic imaging overcomes the resolution limitations of optical techniques. OAI agents like OMV^{Mel}, generates ultrasound after light excitation. Therefore, developing OMV^{Mel} for optoacoustic

applications is a paradigm shift in imaging and theranostic performance compared to previous work based on fluorescence imaging. We have explained better these aspects in:

- V Ntziachristos Nature Methods 7 (8), 603 2010 or
- Adrian Taruttis, V Ntziachristos Nature Photonics 9 (4), 219 2015

Different from fluorescent theranostics that have two competing processes (fluorescence vs nonradioactive decay), both OA signal and photothermal therapy (PTT) effects are related to photothermal conversion efficiency, making OAI/ PTT theranostics easier to design in principle. The strong NIR absorption and photo-thermal conversion capability of OMV^{Mel} makes them highly promising for combined OAI and PTT application.

For these reasons, the work presented in our manuscript brings a new dimension in the research and development of biological nanocarriers (OMVs and extracellular vesicles) for imaging and theranostic applications, in particular in relation to enhancing contrast in optoacoustic imaging and enabling more accurate monitoring of OMV bio-distribution and hyperthermia effects compared to fluorescence carriers.

2. The bacterial outer membrane vesicles (OMVs) can cause immune response in biological environment and the OMVs are less toxic than the bacteria itself but may be toxic. Did the developed OMV^{Mel} show no immune response? If not, did you make any special treatment to stop the immune response?

Response 2: We share the concerns of the reviewer that OMVs can cause immune response. For this reason, in the present manuscript, we used an *E. coli* strain previously modified to be less endotoxic due to mutations in *msbB* gene. This gene is responsible for biosynthesis of lipopolysaccharides (LPS). The *msbB* mutations used have been shown to prevent severe immune response in mice receiving OMVs such as the ones employed in this study, since they result in expression of less endotoxic or inactivated LPS {Gujrati, Vipul, et al. ACS nano 8.2 (2014): 1525-1537}. Therefore no special treatment was required to inhibit systemic toxicity in the mice in the present work.

Nevertheless, to confirm the *in vivo* safety of OMVs, we performed new experiments to investigate whether single dose systemic injection of PBS or OMVs (OMV^{WT}, OMV ^{Δ msbB}, or OMV^{Mel}) would stimulate the immune system in C57BL/6 mice (please see revised text in Page 11, 14 and 15 of revised manuscript). Levels of the cytokines TNF- α , IL-6, and IFN- γ in serum were measured by ELISA. Cytokine levels were evaluated at 2 and 24 h in order to monitor early and delayed immune responses. All three types of OMVs increased serum levels of the three cytokines at 2 h, with OMV^{WT} triggering the greatest increases. In all cases, cytokine levels decreased close to baseline by 24 h. In addition, histology of heart, liver, spleen and kidney at 24 h after injection did not indicate significant organ damage under these treatment conditions. These results suggest that the underacylated LPS on OMV ^{Δ msbB} and OMV^{Mel} induce milder systemic inflammation than the intact LPS on OMV^{WT}, and that the modified OMVs are well tolerated upon systemic administration.

3. How is OMV^{Mel} released from the body after it has accumulated in the cancer? Does OMV^{Mel} accumulate in other organs, such as the liver or kidneys? It would be nice if the authors could add photoacoustic images of other organs after OMV^{Mel} injection.

Response 3: We performed new experiments to analyze the biodistribution of OMV^{Mel} in tumour, a tumour-adjacent area, liver and kidney using MSOT imaging (please see page no. 9 and Supporting Figure 1). The biodistribution of OMV^{Mel} in a 4T1 tumour mouse was monitored continuously during 120 min after intravenous injection. 2h hours monitoring ensured observations without moving the animal from the examination bed to improve the accuracy of the observation. 2h is the maximum amount of time we could maintain the animal under anaesthesia due to our ethical guidelines. During this 2-h period, the MSOT signal gradually increased in the tumour, tumour-adjacent region just below the skin as well as in liver and kidney (please see Supplementary Fig. 1a). These measurements provided further evidence that melanin is the primary source of contrast in our MSOT set-up, and it suggests that OMV^{Mel} circulates and distributes in various organs. At the same time, the melanin signal in tumours rose consistently even from early time points (please see Supplementary Fig. 1b) and our experiments at 3 and 24 h (please see Fig. 4a) showed appreciable signal in tumours over the extended time periods. This persistent signal in the tumour suggests passive tumour targeting ability of OMV^{Mel}, likely due to its nanometre size, which ensures penetration through the leaky blood vessels, as well as due to EPR effects in the tumour region.

In the revised manuscript we also now discuss how previous studies have suggested that OMVs are degraded in late endosomes or lysosomes, and that intracellular melanin is broken down by NADPH-dependent oxidoreductase (see page 15 of the manuscript).

4. In Measurement of photothermal activity section, there is a typo. “Tunable optical parameteric oscillator laser ◇ Tunable optical parametric oscillator laser”

5. A scale bar should be added in Figure 4. (a).

6. Make Y-axis scale of figure 4. (c) and (d) the same for a better comparison.

Responses 4-6: Thanks for providing meticulous comments on our manuscript. These changes have been made.

Reviewer #2 (Remarks to the Author):

This study demonstrates the use of bioengineered vesicles as a potent agent for optoacoustic imaging, with the potential to enable both image enhancement and hyperthermia applications. This manuscript was well prepared and the results was also described well. However, there is a pity in the contents of the study results. This paper emphasizes the use of bioengineered vesicle containing melanin as a theranostic agent. Therefore, the authors should propose results for cancer therapy in vivo by hyperthermia. In addition, the experimental method needs to be presented in more detail. For instance, the method for the measurement of melanin amount in the tumor tissue should be described. Unfortunately, I do not recommend publishing this manuscript.

Response: We thank the reviewer for taking time to provide such insightful comment on our work. These comments greatly motivated us to improve the work. As suggested, we have performed *in vivo* photothermal experiments that support the therapeutic potential of OMV^{Mel} (please see page 9, 10, 13 and 23). We also describe in greater detail optoacoustic melanin measurements (see page 20 and 21).

Reviewer #3 (Remarks to the Author):

This study describes an interesting and novel application for bacterial outer membrane vesicles:

delivery of melanin chromophore for optoacoustic imaging and therapy applications. An ingenious method for OMV incorporation is used, and the described results provide a first proof of principle for this approach.

Response: We thank the reviewer for the insightful discussion and suggestions, responses to the raised queries are discussed below in a point-to-point format.

I have the following questions about the results:

1. Figure 3 shows temperature curves for OMVs (Mel) but not for the empty OMVs. It would be helpful to include this control as well. Similarly, panel 3e shows the cytotoxic effect after irradiation for the OMVs (Mel) but lacks the control without melanin.

Response 1: Data for these controls have been added.

2. If I understand Fig.4 correctly, it only shows data for tumor tissue. A comparison with other tissues is necessary to validate the targeting specificity.

Response 2: Thank you for your comment, similar concerns were raised by Reviewer-1 (Q3). As suggested, we now report the *in vivo* distribution of OMV^{Mel} monitored using multispectral optoacoustic tomography (MSOT). Results show time dependent distribution of OMV^{Mel} in tumour and other tissues (just below the skin) and organs (observed in liver and kidney). The consistent rise in melanin signal at early time points and high signals even after 24 h from tumor indicates passive targeting ability of OMV^{Mel}, which is mainly due to nanometre size of OMVs that assures penetration through the leaky blood vessels and EPR effects in tumour region. (Please see revised text in Page 9 of revised manuscript and Supporting Figure 1)

More general remarks:

1. I am not familiar with the term "theranostics". Please clarify.

Response 1: We provide a brief definition of this term on page 4 and 5.

2. Melanin is a self-antigen. It is possible that presentation through OMVs will lead to immunogenicity of melanin, as OMVs are known to be an efficient vaccine delivery system. So there is a theoretical risk of breaking tolerance and inducing autoimmunity. This may become a problem when clinical application is considered. Please discuss this issue.

Response 2: Similar concerns were raised by Reviewer-1 (Q2). We carried out *in vivo* immunogenicity test for OMVs and discussed this issue to help guide further work on this area (Please see revised text in Page 11 of revised manuscript and Supporting Figure 2).

REVIEWERS' COMMENTS:

Reviewer #1 (Remarks to the Author):

This manuscript, "Bioengineered bacterial vesicles: a novel class of biological nano-heaters for optoacoustic imaging", developed bacterial outer membrane vesicles (OMVs) encapsulating biopolymer-melanin for tumor targeting and photoacoustic imaging contrast enhancement. The authors addressed all the comments well including immune response issue, body accumulation, and advantage of combining therapy and optoacoustic. Therefore, we recommend this work to be published in Nature Communication.

Reviewer #3 (Remarks to the Author):

The authors have satisfactorily addressed the points I raised previously.

Reviewers' comments:

Reviewer #1 (Remarks to the Author):

This manuscript, "Bioengineered bacterial vesicles: a novel class of biological nano-heaters for optoacoustic imaging", developed bacterial outer membrane vesicles (OMVs) encapsulating biopolymer-melanin for tumor targeting and photoacoustic imaging contrast enhancement. The authors addressed all the comments well including immune response issue, body accumulation, and advantage of combining therapy and optoacoustic. Therefore, we recommend this work to be published in Nature Communication.

Response: We thank the reviewer for appreciating the work and recommending the publication in Nature Communication.

Reviewer #3 (Remarks to the Author):

The authors have satisfactorily addressed the points I raised previously.

Response: We thank the reviewer for appreciating the work and recommending the publication in Nature Communication.